# Evaluating Headache and Facial Pain in a Headache Diagnostic Laboratory: Experiences from the Danish Headache Center

**DOI:** 10.3390/diagnostics13162671

**Published:** 2023-08-14

**Authors:** Henrik Winter Schytz, Jeppe Hvedstrup

**Affiliations:** 1Department of Neurology, Danish Headache Center, Rigshospitalet Glostrup, Faculty of Health and Medical Sciences, University of Copenhagen, Valdemar Hansen Vej 5, DK-2600 Glostrup, Denmark; jeppe.hvedstrup.mann@regionh.dk; 2Department of Clinical Medicine, Faculty of Health and Medical Sciences, University of Copenhagen, Blegdamsvej 3B, DK-2200 Copenhagen, Denmark

**Keywords:** primary headaches, secondary headaches, diagnostic testing, headache laboratory, quantitative sensory testing, ultrasound, blink reflex, human provocation models

## Abstract

Background: Diagnostic tests are not routinely used for the diagnosis of primary headaches. It is possible that laboratory tests could be developed and implemented at tertiary headache centers to be an integrated part of the diagnosis and management of headache patients, and laboratory tests that can be used on-site at headache centers could help in evaluating patients with secondary headache disorders. Methods: In this narrative review, we present some of the studies that have been made so far at the Headache Diagnostic Laboratory at the Danish Headache Center that aim to investigate and phenotype primary headaches and investigate secondary headaches as well as improve management. Results: Semi-structured interviews and deep phenotyping, quantitative sensory testing, and provocation studies have been shown to be valuable in categorizing primary and secondary headache subtypes, possible pathophysiology, and defining needs for further research. In patients suspected of increased intracranial pressure, transorbital ultrasound with measurement of the optic sheath diameter may be useful in monitoring patients. The management of headache patients needs to be critically evaluated to optimize treatment continuously. Conclusion: A Headache Diagnostic Laboratory is very useful and should be an integrated part of headache care and management at tertiary headache centers.

## 1. Introduction

Primary and secondary headaches are diagnosed based on the International Classification of Headache Disorders [1]. The classification system is very useful for classifying headache disorders in a systematic fashion based on the systematic description of headache and accompanying symptoms. To exclude patients with symptoms of a primary headache that actually have an underlying secondary headache, it is also important to identify red flags in the patient’s history, clinical findings on the neurological exam, or imaging findings [2]. However, there are still no established laboratory tests [3], except the indomethacin test, used in the ICHD classification system to diagnose and subclassify primary headaches [1]. This review presents studies performed at the Danish Headache Center that can be used at other tertiary headache centers to assist in the diagnosis and evaluation of headache and facial pain.

## 2. Methods

At the Danish Headache Center, we have since 2018 developed a Headache Diagnostic Laboratory (HDL) to investigate and develop methods that serve as useful tools in the diagnosis and classification of both primary and secondary headaches. The HDL is located in the outpatient clinic, where more than 5000 headache patients are being treated and investigated, and which receives 1600 new headache patients per year [4], and can therefore serve to investigate and follow patients managed at the Danish Headache Center. The core staff consist of a consultant in neurology, a nurse and a research laboratory technician trained in headache clinical work and research, and a secretary. The consultant in neurology plans and conducts studies in collaboration with the nurse and laboratory technician as well as medical and PhD students. In this narrative review, we present studies that have been conducted at the HDL to date and identify the challenges and opportunities of having a headache diagnostic laboratory at a tertiary headache center.

## 3. Neurophysiological Studies—The Blink Reflex

Neurophysiological tests could potentially have a role in a headache diagnostic laboratory due to direct testing of cranial nerves possibly involved in headache or facial pain, as well as being inexpensive and easy to use. However, no studies have so far shown a clear clinical role for the use of, for example, the blink reflex in headache [5] or facial pain diagnosis or management [6]. To date, no studies have directly investigated whether the blink reflex responses differ between patients with either classical or idiopathic trigeminal neuralgia (TN). If a blink reflex, which can be conducted in less than 15 min on-site at the HDL, can help distinguish between subgroups of TN patients, this would provide an important tool for fast-tracking patients to the proper treatment. We, therefore, aimed to investigate differences in blink reflex responses between classical and idiopathic TN patients and conducted a study with 55 TN patients [7]. The lab technician and nurse performing and recording the blink reflex were blinded toward the TN diagnosis and site of pain. The study demonstrated, surprisingly, absolutely no differences between the two TN groups [7]. Thus, we could not demonstrate that the blink reflex would be useful in the differential diagnosis of classical versus idiopathic TN.

## 4. Headache Phenotyping Using Quantitative Sensory Testing

Phenotyping studies are important in clinical headache research to further develop headache classification and treatment strategies, as well as our knowledge of the underlying pathophysiology in headache subtypes. Thus, even though migraine diagnoses are clearly defined, it is likely that types of migraine attacks and patients are still quite heterogeneous. Thus, patients may vary based on prodromal, ictal, and postictal symptoms [8,9]. These features might be used to subgroup patients with certain migraine attacks. If subgroups are also identified by specific differences in quantitative sensory testing or other laboratory tests [3], this could imply differences in underlying pathophysiology, and divisions into subgroups might be used to develop novel individualized treatment strategies.

To examine subgroups of migraine patients from a tertiary headache center, a phenotyping study of 100 migraine patients and 100 gender- and age-matched controls was performed [10]. The goal was to map differences between migraine and controls [10], but also differences among migraine patients based on migraine frequency and the presence of ictal neck pain [10]. All individuals had a large test battery performed consisting of questionnaires, semi-structured interviews, and examinations of muscle tenderness examined with total tenderness score (TTS) and local tenderness examination (LTS) [10]. Allodynia was examined using cold pain threshold (CPT) and heat pain threshold (HPT). All examinations were performed interictally (in the absence of a migraine attack). The total tenderness score (TTS) examines eight different pericranial and insertions bilaterally. A subdivision of the score can be made according to their sensory innervation (trigeminal or cervical), i.e., the cephalic-TS (trigeminal innervation) and neck-TS (cervical innervation). Muscle tenderness has previously been demonstrated in both tension-type headache (TTH) [11,12,13] and migraine [11,14,15], but muscle tenderness within different migraine phenotypes has not been investigated systematically before. The study showed that migraine patients had increased muscle tenderness compared with controls and that the increased pericranial tenderness was generalized both in the trigeminal and cervical innervated muscles [10]. TTS was higher in chronic migraine patients compared to episodic migraine patients. A positive association with headache frequency, as well as allodynia, was found in migraine patients. For each monthly headache day, the TTS was increased by 0.34 (CI: 0.17–0.51, *p* < 0.001). The cold pain threshold and heat pain threshold were measured using a TSA-II thermode. Testing was performed in accordance with the German Research Network on Neuropathic pain [16,17] and thresholds were used to define patients as being allodynic [18]. Data showed that having allodynia was associated with a higher total tenderness score of 3.36 (0.35–6.36) [10], which demonstrates that both tests indicate sensitization at a peripheral or central level outside of migraine attacks. The semi-structured interview of the 100 migraine patients also showed that 52% had ictal neck pain, which was defined as neck pain or stiffness 0–48 h before or during the migraine attack, or 24 h after the attack. In patients with ictal neck pain, the TTS was increased. The signal was driven by a difference in neck-TS, and no difference was found in cephalic-TS. Local tenderness score was examined using the palpometer, which is an instrument that measures the elicited pressure and thus standardizes the pressure, a process that is also known as pressure-controlled palpation [19]. Using the local tenderness score, we found a higher tenderness in the proximal part of the trapezius, but not in the distal part of the trapezius, in patients with ictal neck pain compared to migraine patients without ictal neck pain [10]. This indicates a localized sensitization and thus probably has a peripheral cause, since a central sensitization would probably be generalized to all the pericranial muscles because of the innervation of the trigeminocervical complex. Interestingly, patients were also asked to report headaches or migraine occurring the week following the examination. It was found that an impending migraine attack was associated with an increased cephalic-TS, but not neck-TS [10]. This raises an interesting possibility of giving patients a method of detecting a high risk of an impending migraine attack, and thus start treatment earlier or plan accordingly.

Pericranial tenderness and pain thresholds studies can illuminate differences between different headache populations. At our laboratory, we have gained a lot of experience with the examination of pressure pain threshold and the total tenderness score in both primary and secondary headache types. Decreased pain thresholds in post-traumatic headache had been shown in a few studies [20,21], but a demonstration of this had not been done with a large cohort. A recent study compared 100 patients with persistent post-traumatic headache after a mild TBI with 100 healthy age- and gender-matched controls [22]. The tenderness of the pericranial muscles and pressure pain thresholds at m. temporalis and m. trapezius (upper and middle part) were examined in both groups. The study showed that patients were found with higher TTS than controls indicating a higher degree of tenderness in pericranial muscles [22]. The PPTs at all measured areas were lower in patients than in controls, which also suggested an increased sensitivity to pressure [22]. The increased sensitivity in myofascial tissue may, therefore, in part contribute to the generation of persistent headache and pain in post-traumatic headache. The cause of the sensitization is unknown, but it has been speculated that it may be caused by peripheral sensitization of nociceptors in the myofascial tissue or sensitization of second-order neurons at the trigeminocervical complex [23].

## 5. Ultrasound Investigations of Pericranial Muscles

To further investigate pathophysiological differences between migraine patients with and without ictal neck pain, pericranial muscle stiffness has also been evaluated using shear wave ultrasound elastography at the HDL [24]. Shear wave elastography uses the principle of measuring the speed of shear waves in tissue. The ultrasound probe emits a push pulse, which sets the tissue in motion and causes shear waves. The speed of the shear waves corresponds to the stiffness of the tissue [25]. In the study, we measured the speed of the shear waves parallel to the muscle fibers of the trapezius muscle in a combined measurement of the splenius capitis and splenius capitis [24]. The method has been tested at the headache laboratory and shown to be reliable [26]. Examination of 100 migraine patients, of whom 52 had ictal neck pain, showed that the patients with ictal neck pain had increased neck muscle stiffness compared to patients without ictal neck pain (mean difference 0.48 m/s, 95% CI 0.08–0.87, *p* = 0.018) and 46 gender- and age-matched controls (mean difference 0.40 m/s, 95% CI: 0.03–0.78, *p* = 0.036) [24]. The areas examined with shear wave elastography were also examined with pressure pain threshold (PPT). To measure PPT, the pressure is slowly increased in a uniform manner, and the subject is instructed to let the examiner know when the sensation of pressure changes to pain. The pressure pain threshold is often lowered in patients with central sensitization. There were no differences between groups in PPT in this study, indicating that the groups are similar centrally and that the difference is probably peripheral [24]. These studies examining patients with ictal neck pain [10,24] indicate that migraine patients with ictal neck pain are different from those without ictal neck pain and that this difference probably is caused by peripheral changes.

## 6. Headache Provocation Models

A method for future subclassifying headache disorders may be the headache provocation model. Using staff who are experienced in the method is critical, making clinical tests and research studies comparable over time. The model has also been used to examine signaling pathways in headache pathophysiology and has been used at the Danish Headache Center to study migraine pathophysiology, but has also proven successful in investigating other areas of the headache field [3,27]. A study performed at the HDL examined patients who had a history of persistent post-traumatic headache for more than 12 months after a mild traumatic brain injury (TBI) [28]. A total of 60 patients completed the non-randomized single-arm open-label study. All patients received an infusion of calcitonin-gene-related peptide (CGRP) for 20 min and completed a headache diary until 12 h after infusion. In total, 43 (72%) had a migraine-like headache during the 12 h period, which implies a role of CGRP in generating migraine-like headache in post-traumatic headache [28]. It can be argued that a pronounced nocebo response may have influenced the study results. However, a double-blinded randomized, placebo-controlled two-way crossover study has also been conducted and showed that CGRP does induce migraine-like headache in patients with post-traumatic headache after a mild traumatic brain injury [29]. Similar to the prior study, patients had had persistent post-traumatic headache for at least 12 months after a mild TBI. However, in contrast, patients were examined on two study days and were randomized to either placebo or CGRP infusion at their first visit. Headache data including phenotype and severity was collected for the following 12 h. A total of 30 patients completed the study. Interestingly, 21 (70%) patients developed a migraine-like attack during the observational period after CGRP infusion, while only 6 (20%) did after the placebo infusion [29]. In addition, headache intensity showed a higher area under the curve after CGRP infusion compared to placebo. The high induction rate indicates the importance of CGRP in the pathophysiology of persistent post-traumatic headache, but the clinical implications of the current findings need to be studied further in detail.

The provocation model is also useful for examining possible differences in migraine subgroups by examining the migraine induction rate. It has been shown that CGRP induces migraine-like attacks in patients with episodic migraine [30]. However, the sensitivity of patients with chronic migraine to CGRP had not been investigated. Thus, a single-arm open-label study was performed to examine the induction of migraine-like headache in patients with chronic migraine [31]. Additionally, the influence of headache on the day of examination and headache frequency for the last month were examined. Patients received an infusion of CGRP over 20 min and were under observation for the next 12 h. A comparison with historical provocation studies of episodic migraine patients was also made using a systematic review. The study included 58 chronic migraine patients with either headache or without headache on the day of examination [31]. In total, 92% of patients with headache on the day of examination and 65% of migraine patients without headache on the day of examination reported migraine-like attacks during the study, which is a significant difference [31]. The induction rate of chronic migraine patients without headache was comparable to historical data of patients with episodic migraine (62%). The association between headache frequency in the last month and migraine-like induction rate was not significant. Thus, the study showed that chronic migraine patients with ongoing headache are more sensitive to CGRP as a migraine trigger.

Future provocation studies may help predict treatment response to CGRP monoclonal antibody treatment, but also to possible future targets for migraine treatments (e.g., PAC_−1_ receptor/PACAP, VIP, or others) [32].

## 7. Semi-Structured Interviews

Features of distinct headache disorders are important for clinical work-up and treatment [33]. Previous research has established that PTH is frequent and associated with high disability, but the patient population still needs to be characterized in-depth [34]. Deep phenotyping of the post-traumatic headache population was therefore performed examining 100 patients with persistent PTH acquired after a mild TBI [35]. Patients were examined with a semi-structured interview and the 12-item allodynia symptom checklist. In the 100 PTH patients, the mean headache frequency was 25.4 ± 7.1, indicating a large headache burden in this population [35]. Most patients (61%) reported a chronic migraine-like headache, whereas the second most frequently reported type was combined episodic migraine-like headache and tension-type-like headache, which was found in 29% [35]. Cutaneous allodynia was not reported in 54%, in a mild manner in 23%, moderate in 17%, and severe in 6% [35]. Interestingly, 63 patients reported preventive headache treatment, but 79% of those who had tried preventive treatments reported failure of at least one preventive medication [35], showing the difficulties in treating post-traumatic headache. This was also emphasized by 19% having failed at least four drugs [35]. Another interesting finding was that 39% of patients had reported triptan use, but close to half of these patients reported no efficacy [35]. Thus, although the headache phenotype is similar in migraine and persistent post-traumatic headache, the findings may indicate differences in underlying pathophysiology and an unmet need for specific treatment strategies in post-traumatic headache. During the semi-structured interviews, patients are asked thoroughly about their history, pain patterns, and the extent of their disability and asked to elaborate on various aspects of their disorder. Phenotyping may in many aspects be the closest thing to biomarkers we have in headache medicine. The interviews and phenotyping resemble narrative-based medicine, which is believed to improve the doctor–patient relationship among other positive effects [36].

## 8. Laboratory Investigations in Intracranial Hypertension

The Danish Headache Center yearly receives 150 new patients with elevated intracranial pressure (ICP) without clear brain pathologies [37,38], which is defined as pseudotumor cerebri syndrome (PTCS). PTCS often occurs as idiopathic intracranial hypertension (IIH) and may also be caused as a complication to several conditions, including side effects [36]. PTCS is defined as the presence of papilledema and increased lumbar puncture opening pressure ≥ 25 cm H_2_O, and normal neuroimaging, CSF contents, and neurological examination [37]. The diagnosis is challenging as papilledema is difficult to assess by neurologists, and overdiagnosis has been reported to occur in up to 40% of cases [39].

Transorbital sonography (TOS), an ultrasound imaging technique, is a safe and non-invasive test that can be used to indirectly assess ICP and papilledema [40,41]. Since CSF around the optic nerves is directly connected with the subarachnoid space surrounding the brain, the optic nerve sheath diameter has been hypothesized to fluctuate with ICP [42]. Based on this, Korsbæk et al. [43] performed a prospective case-control study at the Danish Headache Center. The patients had new-onset PTCS and were matched with healthy controls. All had fundoscopy, lumbar puncture with opening pressure, and TOS assessed by a blinded observer. The study demonstrated that the optic nerve sheath diameter and optic disc elevation were significantly increased in PTCS patients compared to healthy controls [43]. A cut-off point for optic nerve sheath diameter of 6 mm resulted in a sensitivity of 74% and specificity of 94% for PTCS, while a cut-off point for optic disc elevation of 0.6 mm resulted in a sensitivity of 100% and specificity of 83% for PTCS. The study, therefore, concluded that non-invasive measurements of the optic disc and nerve conducted at the Headache Diagnostic Laboratory can achieve high specificity and excellent sensitivity for PTCS. This diagnostic tool is excellent for following patients with PTCS undergoing treatment and when suspected of relapse.

Papilledema is a hallmark of PTCS, reflecting pathologically increased ICP, which untreated, leads to irreversible vision loss [38]. Therefore, it is of great importance to assess papilledema and visual field loss accurately. However, detecting the presence of papilledema using conventional non-dilated direct ophthalmoscopy, which is common practice by neurologists at tertiary headache centers, is very hard to master and requires considerable practice [44]. In fact, using a direct ophthalmoscope, the field of view is only 5° as opposed to the at least 45° field of view of most fundus photography cameras [44]. At the Headache Diagnostic Laboratory, we have, therefore, introduced a system combining imaging of the optic disc (fundoscopy) and automated visual field perimetry (COMPASS, CMP, CenterVue, Padua, Italy). The system is used to detect papilledema and visual field impairment in patients suspected of or followed for PTCS. To validate the use of the Compass system, we tested the system by obtaining blinded fundus images and perimetry from the COMPASS system in comparison with measurements performed within 7 days at a neuro-ophthalmology outpatient clinic [45]. All images were assessed by a neuro-ophthalmologist. Furthermore, interrater assessments were done by comparing the ratings of a trained neurologist, an untrained medical doctor, and a trained medical student. In regard to papilledema, the inter-method variation showed a kappa value of 0.60, sensitivity of 87%, and specificity of 73%. The rating between the neuro-ophthalmologist and the headache center staff showed kappa values from 0.43 to 0.74, with sensitivity values from 70% to 96% and specificity values from 46% to 93%. The COMPASS system showed moderate agreement and only a 59% sensitivity in detecting visual field defects compared with the neuro-ophthalmology outpatient clinic system, while there was only slight to fair agreement (from 0.19 to 0.31) in visual field assessment between the headache center staff and the neuro-ophthalmologist [45]. The validation study, therefore, demonstrates that clinical staff can use the COMPASS system with reasonable sensitivity in detecting papilledema in patients suspected of or followed with PCTS at a headache center [45].

## 9. Headache Diagnostic Laboratory to Assess Management Strategies

The Headache Diagnostic Laboratory is also involved in monitoring the management strategies at the Danish Headache Center. It is of importance not only for the Danish Headache Center but also for the management of headache at all treatment levels (primary, secondary, and tertiary) to investigate previous non-pharmacological and pharmacological headache treatment and clinical characteristics in newly referred headache patients. This can lead to a change in strategy of how headache management should be offered for the many headache sufferers, who have a high need for optimal patient-centered treatment [36].

We, therefore, performed an observational, cross-sectional study conducted on patients on their first visit to the Danish Headache Center between May 2020 and March 2021 [4]. The analysis comprised 382 patients with a migraine and/or TTH diagnosis. The study showed that new patients consisted of episodic migraine (36%), chronic migraine (43%), episodic tension-type headache (3%), and chronic tension-type headache (17%). The majority of patients had attempted non-pharmacological treatments, e.g., physiotherapy and acupuncture, which shows that patients often are very active in seeking headache treatment. For pharmacological treatment, 71% with episodic migraine and 66% with chronic migraine had tried one triptan or less [4]. Patients that had never tried preventive medication were identified among 35% of episodic migraine, 19% of chronic migraine, 50% of episodic tension-type headache, and 41% of chronic tension-type headache patients [4], which shows that there is likely an unmet need for proper preventive treatment. Similar studies may also be beneficial in other centers. A systematic review found that the most frequent reason for primary care visits in low- and middle-income countries was headache [46]. Examinations of those who are referred may shed light on any low-hanging fruits for headache treatment that can be implemented in primary or secondary care.

Within a tertiary headache center, there is also a great need to optimize treatment to avoid unnecessary waiting lists or patients having to wait unnecessarily long periods for the correct treatment. For many years, the treatment strategy has been to see a new patient for a first visit with a follow-up by an experienced headache specialist 4–6 months later. If a patient is recommended pharmacological treatment at the first visit but is not compliant, this may result in patients being untreated for several months due to a lack of follow-up. We, therefore, conducted a study investigating if telephone follow-up consultations resulted in better adjustment of treatments and a higher degree of patient satisfaction in migraine and TTH patients [44]. The study was conducted as a prospective quality control study with controls receiving business-as-usual treatment [47]. The telephone interview intervention group was contacted by telephone for typically 5–10 min 8 and 16 weeks after their first visit to the headache center. In total, 96 telephone interview intervention patients and 91 business-as-usual patients were included in the analysis [47]. More patients in the telephone interview group than in the business-as-usual group had a change in acute medication and preventive medication. We did not find differences in headache reduction between the groups, but there was a higher degree of patient satisfaction in the telephone interview group compared with the business-as-usual group [47]. The study shows that two short simple telephone follow-ups within the first 6 months of migraine and TTH patient treatment courses result in more efficient treatment and higher patient satisfaction.

## 10. Discussion

In the present review, we have described 12 studies conducted at the HDL. The HDL has so far gathered data that may be useful for the classification and individualized treatment of headache, for the diagnosis of patients with altered intracranial pressure [41,43], and for the management of patients followed at the Danish Headache Center. The methods are now well-tested and protocolized in the laboratory and thus, can be performed routinely (Figure 1).

The HDL is located within the outpatient clinic and has been running smoothly with the daily clinical routines. Some challenges are worth mentioning and are important to acknowledge when running the HDL. First, it may not always be possible to diagnose correctly at the first visit, as patients at a tertiary headache center are often quite complex in symptomatology and presentation. This affects the inclusion of patients, who may need re-evaluation and further work-up. Thus, the staff at the Headache Diagnostic Laboratory need to continuously monitor the patients that are being investigated in different projects. Second, patients do not always have the capacity to be investigated with sensory testing and headache provocation, which may lead to selection bias in the patients included in the studies. Finally, at a tertiary headache center, there are often clinical trials and research studies that are actively recruiting patients, and it is important that all of these activities are performed without unnecessary confusion and disturbance to the patients. Conversely, the opportunities are vast for using a headache diagnostic laboratory, which can serve as a unit for testing clinical hypotheses in a systematic and thorough fashion and monitoring clinical performance.

## 11. Conclusions and Future Prospects

The HDL has shown itself valuable in testing and phenotyping headache patients. This continuing work is important in developing the headache field and exploring future treatment perspectives as shown in migraine and PTH studies. In clinical work, the support of the HDL has been important in describing the referred population and thereby finding possible ways of increasing the effectiveness of patient management not only at the headache center but also in primary care. Additionally, when new equipment and methods need verification, the HDL has shown its worth by providing technical know-how and methodical testing.

At the HDL, we continue to strive to develop new methods and approaches that would be useful in the headache clinic. In future studies, we will investigate if CGRP provocation is useful as a predictor for CGRP monoclonal antibody treatment [48] and if genetic risk score or metabolic profiling can be a predictor for treatment outcome. Other secondary headache disorders, such as spontaneous intracranial hypotension [1] also need to be investigated at the HDL using ultrasound techniques. Rare primary and secondary headaches also need to be phenotyped as well as the effect of off-label treatment. The testing and use of digital diagnostic tools [49,50,51] and headache calendars would also be of interest in future studies. Imaging studies have been useful in illuminating the pathophysiology of migraine and may be a future biomarker for treatment response [52].

We suggest that for developing and implementing diagnostic tests and evaluating clinical characteristics of headache at tertiary headache centers, a headache diagnostic laboratory is a key element. Such a department may be an integrated part of specialized headache centers and can have a similar setup as the HDL at the Danish Headache Center, which consists of a consultant in neurology, a nurse, a research laboratory technician, and a secretary. At present, the subclassification of headache disorders according to characteristics and tests is possible, but the clinical implication is still largely unknown.

## Figures and Tables

**Figure 1 diagnostics-13-02671-f001:**
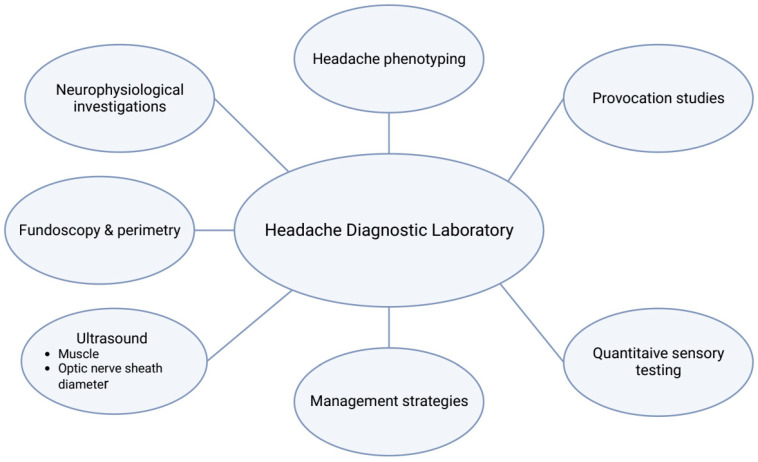
Overview of the different areas of investigations conducted at the Headache Diagnostic Laboratory.

## Data Availability

Not applicable.

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
