# Peer review of "Evaluating Headache and Facial Pain in a Headache Diagnostic Laboratory: Experiences from the Danish Headache Center"

_diagnostics, 2023, doi:10.3390/diagnostics13162671_

Round 1

Reviewer 1 Report

The Authors expertly address an important topic in the management of primary and secondary headaches. However, it would be helpful from the beginning (title) to make it clear to whom this review is addressed. In my opinion to Level III Headache Centers, where care is mixed with research. In fact, the majority of primary headache methods have partial clinical reliability, and cannot be considered as biomarkers for all those who face headaches every day (95% of headache visits are made outside the Headache Centres circuit), partly because these have waiting lists that are not compatible with the necessary turn-over of care. This number is then absolute in the South Globe.

See: Bigio J, MacLean E, Vasquez NA, Huria L, Kohli M, Gore G, et al. (2022) Most common reasons for primary care visits in low- and middle-income countries: A systematic review. PLOS Glob Public Health 2(5): e0000196. https://doi.org/ 10.1371/journal.pgph.0000196

Regarding secondary headaches then the limitation lies in the fact that only certain types of secondary headaches fall within the purview of neurologists (almost 100% of area experts) whereas the IHS classification would provide for skills spread over all medical specialties, a fact that is absolutely unlikely due to the low interest of non-neurologist specialists. Narrative-based Medicine will have to find a place in headache disorders so that we will not have one or more universally usable biomarkers.

I recommend also considering the following references, relative to these topics:

PMID: 37217887

PMID: 37221469 

https://doi.org/10.1007/s42399-020-00556-x

https://doi.org/10.1007/s42399-023-01423-1

PMID: 35690726

Finally, the overall value of the review is commendable if these limitations will be clearly expressed.

Reviewer 2 Report

The paper presented to me for review concerns the organization of the Headache Diagnostic Laboratory at one of the world's top research centers - the Danish Headache Center in Copenhagen. Undoubtedly, it is a treasure trove of knowledge for all centers that deal with headaches.

The work is written in a very understandable clear language, presents all diagnostic aspects and discusses them in detail.

Before accepting the paper for publication, I would suggest one addition: the subsection on experimental models should highlight other potential targets for migraine that are still being sought on the basis of: PMID: 37370051  
